# Reservoir Transformer at Infinite Horizon: the Lyapunov Time and the Butterfly Effect

## Abstract

We introduce Reservoir Transformer with non-linear readout, a novel neural network architecture, designed for long-context multi-variable time series prediction. Capable of efficiently modeling arbitrarily input length sequences, our model is powerful in predicting events in the distant future by retaining comprehensive historical data. Our design of a non-linear readout and group reservoirs overcomes the limitations inherent in conventional chaotic behavior prediction techniques, notably those impeded by challenges of prolonged Lyapunov times and the butterfly effect. Our architecture consistently outperforms state-of-the-art deep neural network (DNN) models, including NLinear, Pyformer, Informer, Autoformer, and the baseline Transformer, with an error reduction of up to -89.43% in various fields such as ETTh, ETTm, and air quality.

## 1 Introduction

Chaos theory offers a profound framework for unveiling the underlying patterns and deterministic laws governing dynamical systems, finding applications spanning various disciplines within human society, such as biology, chemistry, physics, economics, and mathematics, among others (Liu, 2010). A captivating facet of chaos theory is its intriguing concept known as "sensitivity to initial conditions," famously referred to as "the butterfly effect" (Jordan & Smith, 2007). When two initial conditions exhibit only a minor disparity, their divergence over time undergoes exponential amplification, contingent upon the Lyapunov time inherent to the system's dynamics. Consequently, forecasting the distant future in chaotic systems presents formidable challenges primarily rooted in two fundamental issues: (1) the need to account for extensive historical sequences and (2) the pronounced sensitivity to initial conditions.

In the realm of addressing time-series tasks, prior research has explored a multitude of techniques, with a notable emphasis on the utilization of deep neural network models, exemplified by Transformer (Zeng et al., 2022). Nevertheless, an intrinsic constraint associated with the Transformer architecture, originally proposed by (Vaswani et al., 2017), manifests in its quadratic time and memory complexity concerning input length. This inherent limitation imposes a significant hindrance when endeavoring to perform long-term forecasting tasks. Several notable efforts, including UnlimiFormer (Bertsch et al., 2023), Efficient Transformer (Tay et al., 2022), and Reformer (Kitaev et al., 2020), have been made to extend the input length of Transformers. These endeavors primarily hinge on modifications to the attention model itself, often founded on certain assumptions. However, the outcome of these approaches is the extension of the input length to a fixed value, thereby yielding limited adaptability for learning from and predicting arbitrary long sequences. We directly tackle the challenges associated with stable and accurate long-term chaotic prediction, aiming to push the boundaries of existing technology in time series forecasting. Our approach represents an innovative solution tailored to the complex task of extremely long-term chaotic prediction. We integrate reservoir computing into the Transformer framework, offering a novel approach that can effectively model sequences of inputs with arbitrary lengths. Reservoir computing, known for its simplicity and computational efficiency, excels in processing temporal data within the context of chaotic time series prediction (Bollt, 2021). Within our framework, the reservoir plays a pivotal role in transforming sequential inputs into a high-dimensional space, enabling the modeling and utilization of inputs with diverse lengths in deep neural networks (DNNs). This marks a significant departure from heuristic-based assumptions and instead introduces a systematic approach to handle extended input lengths, thereby revolutionizing the field of chaotic prediction. This ensemble consistently captures and re-

tains inputs from all prior time stamps, laying a robust foundation for continuous time stamp learning and enabling efficient long-term context assimilation. Following this, the output from the ensemble seamlessly integrates with the observation of the present time stamp, subsequently being routed to Deep Neural Networks (DNNs), such as the Transformer, facilitating short-term context comprehension. Our goal revolves around refining predictions associated with the current time stamp's value or classification. By employing this method, we efficiently discern long-term time stamp interconnections through the sophisticated training capabilities of the ensemble reservoirs while simultaneously leveraging DNNs for the nuanced learning of contemporaneous features.

Our framework grapples with two technical obstacles related to Lyapunov time and the butterfly effect. Firstly, the confined input length of the Transformer considerably constrains the reservoir's dimensions and potential. This limitation arises since the number of reservoir nodes scales quadratically with the reservoir output size, which mirrors the input size of DNN models, including the Transformer. To efficiently manage longer sequences, our system employs a nonlinear readout rooted in single-head attention mechanisms, superseding the traditional linear reservoir readout. Given its heightened expressiveness, this nonlinear readout essentially performs a dimensionality reduction on reservoir outputs, thus providing the Transformer with more meaningful and potent feature inputs. The second predicament intrinsic to our framework pertains to prediction discrepancies based on varied initializations. To circumvent the sensitivities tied to neural network starting points frequently witnessed in chaotic systems, we incorporate ensemble learning with multiple reservoirs. This strategy notably elevates prediction precision and consistency. Empirical results showcase that the nonlinear readout, when sourced from ensemble reservoirs, drastically refines the Transformer's prowess in time-series predictions, registering an error reduction of up to 89.43%. Through experiments, we gauge the chaotic nature inherent in our time-series datasets. Furthermore, our methodology surpasses leading-edge DNNs, aptly handling diverse input lengths, signaling a transformative phase for the Transformer model.

Our contributions are mainly threefold: a) We integrate reservoir computing into the system, enabling the Transformer to tackle long-term predictions effectively. b) We refine traditional reservoir computing techniques by substituting the linear readout with a nonlinear counterpart, facilitating streamlined dimensionality reduction and adept feature assimilation. c) To address the nuances of reservoir initialization sensitivity, we implement ensemble reservoirs.

## 2 BACKGROUND

Consider a historical time series data containing $\vec{u}(t)$, a vector of features with $c$ components, and $\vec{y}(t)$ is corresponding labels for each time stamp $t \in T$, where $T$ is the total time stamps in a finite history. Now we aim to forecast $\vec{y}(t+1)$ the class or the value of a future time step given $t$ sequential multivariate history $\vec{u}(1:t+1)$ and $\vec{y}(1:t)$.

**Time-Series Forecasting** often only handles fixed input length with an assumption of looking back $s$ window size and considering $(t-s)$ history to predict at $t$ instead of using the complete history. Then the task is to predict $\vec{y}(t+1)$ given $\vec{u}(t-s:t+1), \vec{y}(t-s:t)$ with the conditional distribution:

$$Pr(\vec{y}(t+1)|\vec{u}(t-s:t+1), \vec{y}(t-s:t); \phi) \tag{1}$$

Here $\phi$ indicates learnable parameters shared by all-time steps $T$ to predict the conditional probability. The reason to introduce $s$ is that learning this conditional probability typically depends on $s$. For example, Transformer takes $\mathcal{O}(s^2 \times c)$. The computational cost is quadratic to the $s$.

We introduce deep reservoir computing for time series in chaotic prediction to avoid this costly operation. The advantage of reservoir computing is to preserve all the history for prediction. Therefore, it becomes possible to estimate conditional distribution with any length input:

$$Pr(\vec{y}(t+1)|\vec{u}(1:t+1), \vec{y}(1:t); \phi) \tag{2}$$

If the reservoir has $m$ size of the output vector, the time complexity of the proposed work will be $\mathcal{O}((k+m)^2 \times c)$. Here, the parameter $k$ represents the size of the small look-back window compared to $s$.

## 2.1 Deep Reservoir Computing

Our basic reservoir architecture follows the deep reservoir computing (DRC) (Gallicchio et al., 2017) with the Echo State Network (ESN) model and a trained linear readout. In the basic Leaky Integrator ESN (LI-ESN) model Gallicchio et al. (2017), the state is updated according to the following state transition function:

$$\vec{x}(t) = (1 - a)\vec{x}(t - 1) + \alpha \tanh(\vec{W}_{in}\vec{u}(t) + \boldsymbol{\theta} + \vec{\hat{W}}\vec{x}(t - 1)), \tag{3}$$

where $\vec{x}(t) \in R^n$ denotes the reservoir state at time $t$, $n$ represents the dimensionality of the reservoir state vector. $\vec{W}_{in} \in R^{c \times n}$ is the input-to-reservoir weight matrix, $\boldsymbol{\theta} \in R^n$ is the bias-to-reservoir weight vector, $\vec{\hat{W}} \in R^{n \times n}$ is the recurrent reservoir weight matrix, $\tanh(.)$ is the element-wise applied hyperbolic tangent activation function, and $\alpha \in [0, 1]$ is the leaky parameter, i.e., the decay rate of the nodes. The reservoir parameters are initialized according to the constraints specified by the Echo State Property (ESP) and then are left untrained Gallicchio & Micheli (2017). Accordingly, the weight values in $\vec{W}_{in}$ and in $\boldsymbol{\theta}$ are chosen from a uniform distribution over $[-scale_{in}, scale_{in}]$, where $scale_{in}$ represents an input-scaling parameter. Matrix $\vec{\hat{W}}$ contains randomly selected values from a uniform distribution. It is then adjusted so that the spectral radius of matrix $\vec{\hat{W}} = (1 - \alpha)I + \alpha\vec{\hat{W}}$, denoted as $\rho$, remains below 1 ($\rho < 1$). This process helps ensure stable dynamics in the reservoir system. The reservoir weight $\vec{\hat{W}}$ and input weight $\vec{W}_{in}$ are fixed and random, where each weight is drawn according to Gaussian distribution with parameterized variances.

## 2.2 Linear Readout

As reservoir output, a readout component is used to linearly combine the outputs of all the reservoir units as in a standard ESN. The output of a reservoir at each time step $t$ can be computed as

$$y(t) = \vec{W}_{out}\vec{x}(t)^T + \theta_{out}, \tag{4}$$

where $y(t) \in R^m$ is the linear readout with $m$ dimensionality. $\vec{W}_{out} \in R^{n \times m}$ represents the reservoir-to-readout weight matrix, connecting the reservoir units to the units in the readout. Since only $\vec{W}_{out}$ are trained, the optimization problem boils down to linear regression. Training is typically not a limiting factor in DRC, in sharp contrast with other neural network architectures. The expressiveness and power of reservoir computing rather lie in the high-dimensional non-linear dynamics of the reservoir.

The reservoir is effective for managing lengthy inputs. When we combine it with a Transformer model, the time complexity becomes $\mathcal{O}((k + m)^2 \times c)$. For the Transformer, if we include more time steps in the input, the training time increases quadratically with the input length. On the other hand, the reservoir treats each time step individually, keeping the entire history in its learning process without making the input longer. This makes the reservoir a better choice for handling long sequences of input data.

## 2.3 Baseline Transformer

In Transformer, the architecture unravels the temporal dependencies into individual time steps (Wu et al., 2020). At each time of training, we optimize $\phi$ parameters by a Transformer $\mathcal{M}$ to learn the distribution $Pr(\vec{y}(t + 1)|\vec{u}(t - s : t + 1), \vec{y}(t - s : t))$ composed of two inputs: the current time stamp $\vec{u}(t + 1)$ and that of the previous $s$ time steps $\vec{u}(t - s : t), \vec{y}(t - s : t)$:

$$\vec{g}(t + 1) = \kappa\vec{u}(t + 1) + (1 - \kappa)(\vec{u}(t - s : t), \vec{y}(t - s : t)) \tag{5}$$

Here $\kappa$ is a learning parameter adjusting the weights of the current input and previous inputs. Thus $\vec{g}(t+1)$ is the input to Transformer, $\vec{\bar{y}}(t+1) = \mathcal{M}(\vec{g}(t+1))$. Nonetheless, employing $\vec{g}(t+1)$ as the input for the Transformer poses two significant challenges. Firstly, the Transformer's input length is constrained by a factorization approach involving a retrospective window of size $s$, thereby increasing computational complexity. Secondly, the Transformer imposes restrictions on input length, preventing it from exceeding the value of $s$ due to quadratic time complexity considerations. Consequently, the incorporation of extensive historical data becomes impractical, limiting its potential impact on the learning and decision-making processes.

## 3 METHOD

### 3.1 READOUT WITH SELF-ATTENTION

One downside of the reservoir is that if the reservoir output size $m$ is large, the Transformer training is slow due to its quadratic complexity. Since a linear readout of a reservoir needs a much larger size to have the same expressive power as the nonlinear readout, we introduce nonlinear readout layers, which work quite well in reducing DRC output dimension and improving prediction performance. Replacing linear readout with nonlinear readout leads to faster convergence and reduces the possibility of getting stuck in a local optimum (Triefenbach & Martens, 2011).

To model nonlinear readout, we use attention mechanisms (Vaswani et al., 2017) to capture input feature importance, alleviating the problem of vanishing gradient of long-distance dependency. There have been various implementations of attention mechanisms. We implement our attention model as in (Zhao, 2022). Attention mechanisms process sequential data that considers the context for each timestamp. $W$ and $b$ are this representation's weight and bias, and the hyperbolic tangent function $\tanh(.)$ is a nonlinear activation.

$$h_{i,j} = \tanh(\vec{x}(t)_i^T \vec{W}_t + \vec{x}(t)_j^T \vec{W}_x + b_i) \tag{6}$$

$$e_{i,j} = \sigma(\vec{W}_\alpha h_{i,j} + b_\alpha) \tag{7}$$

$$\gamma_{i,j} = \frac{exp(e_{i,j})}{\sum_{i=1}^{J}(exp(e_{i,j}))} \tag{8}$$

$$r(t) = \left\{ \sum_j \gamma_{i,j} \vec{x}(t)_j \right\}_{i=1}^{c} \tag{9}$$

Here, $r(t)$ is a non-linear readout using the attention mechanism and the subscript $i$ and $j$ are the $i$-th and $j$-th components of $\vec{x}(t)$. The attention weight $\gamma$ is a softmax of $e$.

### 3.2 GROUP RESERVOIR

In chaos theory, the butterfly effect is the sensitive dependence on initial conditions in which a small change in one state of a deterministic nonlinear system can result in large differences in a later state. To enhance reservoir performance, we integrate multiple reservoirs' nonlinear readout layers, ensuring their independence. We consider $Q$ reservoirs with distinct decay rates ($\alpha$ and $\rho$) initialized randomly Gallicchio et al. (2017). Utilizing Equations 3 and 9, if $r^l(t)$ represents the nonlinear readout from the attention mechanism of the $l$-th reservoir within $Q$, we combine various leaky parameters ($\alpha$) and spectral radius ($\rho$) across the reservoirs. This results $\vec{o}(t)$ in a grouped ESN representation in element-wise $\oplus$ addition to all reservoirs' attention outputs:

$$\vec{o}(t) = r^1(t) \oplus r^2(t) \oplus ... \oplus r^Q(t) \tag{10}$$

### 3.3 RESERVOIR TRANSFORMER (RT)

Our group reservoir efficiently models observed time-series data in dynamical systems and require minimal training data and computing resources (Gauthier et al., 2021) and is ideal for handling long sequential inputs. Here we mainly describe adopting a reservoir for Transformer Vaswani et al. (2017) as one of the most well-known and valuable DNN architectures. However, our methods are adaptable to other DNN model architectures, for whom we also compared results in our experiments.

Figure 1 shows the architecture of our reservoir-enhanced Transformer. The left figure shows the ensemble of reservoirs, and the right figure shows how to use a non-linear readout reservoir to help Transformer handle arbitrarily long inputs.

Our approach enables comprehensive modeling of entire historical data in decision-making. While the Transformer captures temporal dependencies (Equation 1), limited by a window of size $s$, our method incorporates reservoir pooling to overcome this limitation. This permits the consideration of the entire history without a rise in time complexity. By incorporating Equation 10 (ensemble of all $Q$ reservoir readouts), we arrive at:

$$\vec{z}(t+1) = \kappa \vec{u}(t+1) + (1-\kappa)\vec{o}(t) \tag{11}$$

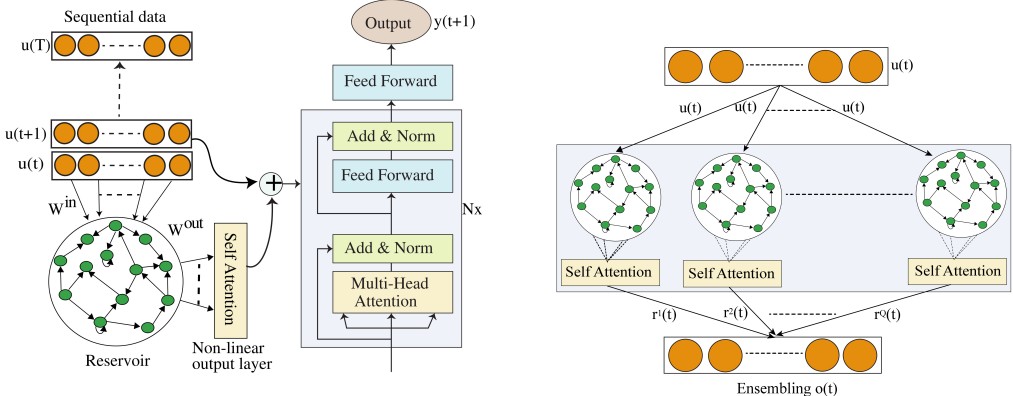

**Figure a**: Non-linear readout for RT.          **Figure b**: Ensemble of multiple reservoirs

Figure 1: Description of deep Reservoir Transformer: In Figure (a), the readout obtained from self-attention, combined with the present input $\vec{u}(t + 1)$, serves as the input for training the transformer architecture. In Figure (b), the input $\vec{u}(t)$ is processed individually by $Q$ reservoirs. The nonlinear output of each reservoir is later combined to form the ultimate output, labeled as $\vec{o}(t)$.

$$\bar{\vec{y}}(t + 1) = \mathcal{M}(\vec{z}(t + 1)) \tag{12}$$

$\vec{z}(t+1)$ denotes the Transformer $\mathcal{M}$ input with $\kappa$ as a learnable parameter indicating the weights of current multivariate states and reservoir state to predict $\bar{\vec{y}}(t + 1)$ by Equation 12.

In the context of multivariate time series prediction, the target label for forecasting up to $q$ future time steps is denoted as $\vec{u}(t+1 : t+1+q; v)$, which is contingent upon the observed series $\vec{u}(1 : t)$. In the multivariate case, all available features are taken into account with the conditional distribution $Pr(\vec{u}(t + 1 : t + 1 + q)|\vec{u}(1 : t); \phi)$.

## 3.4 TRAINING

We enhance the non-linear readout layers through self-attention mechanisms within ensemble reservoirs. This involves updating these layers and performing element-wise addition of the readouts, resulting in an output denoted as $\vec{o}(t)$. This output is concatenated with the present input features $\vec{u}(t+1)$, yielding $z(t+1)$, which then undergoes processing via the Transformer encoder $M(.)$. Initially, we sequentially train the model $M(.)$ across all time stamps $1 : T$ using the training dataset. Subsequently, we fine-tune our models using the validation set to achieve optimal model performance. The model is subjected to testing with varying hyper-parameters such as reservoir size, leaky rate, spectral radius, number of reservoirs, learning rate, attention size, Transformer block, and dropout rate. The choice of these hyper-parameters may differ depending on the specific dataset employed (details in the appendix). For a comprehensive understanding of our training algorithm, refer to the pseudo-code provided in the appendix, along with a detailed explanation of the notation employed. For **regression** tasks, we use the Huber loss function Meyer (2021) as our objective (Equation 13) and for **classification** tasks, we use cross-entropy loss for $K$ classes (Equation 14):

$$\mathcal{L}(\vec{y}, \bar{\vec{y}}) = \frac{1}{T} \sum_{i=1}^{T} \begin{cases} \frac{1}{2}(\vec{y}_i - \bar{\vec{y}}_i)^2, & \text{if } |\vec{y}_i - \bar{\vec{y}}| \leq \delta \\ \delta|\vec{y}_i - \bar{\vec{y}}_i| - \frac{1}{2}\delta, & \text{otherwise,} \end{cases} \tag{13}$$

$$\mathcal{L}(\vec{y}, \bar{\vec{y}}) = \frac{1}{T} \sum_{i=1}^{T} \sum_{j=1}^{K} \vec{y}_{i,j} \log \bar{\vec{y}}_{i,j} \tag{14}$$

Here $\delta$ is an adjustable parameter that controls where the function change occurs to keep the function differentiable, $T$ is the total number of samples and $\vec{y}, \bar{\vec{y}}$ are ground truth and predicted values respectively. In the same way, for time series

# 4 EXPERIMENTS

We show our experimental results of Reservoir Transformer (RT) compared to baselines, including state-of-the-art methods in time-series prediction, such as: Nlinear (Zeng et al., 2022), FEDformer (Zhou et al., 2022), Autoformer (Wu et al., 2021), Informer (Zhou et al., 2021), Pyraformer (Liu et al., 2021), LogTrans (Li et al., 2019), GRIN (Cini et al., 2021), BRITS (Cao et al., 2018), STMVL (Yi et al., 2016), M-RNN (Yoon et al., 2018), ImputeTS (Moritz & Bartz-Beielstein, 2017), and Transformer Vaswani et al. (2017).

## 4.1 RESULT ANALYSIS

We thoroughly assess our RT method on different time series regression tasks. These tasks include **Electricity**, **Traffic**, **Weather**, **ETTh1**, **ETTh2**, **ETTm1**, **ETTm2**, **ILI**, **Exchange Rate** (Zhou et al., 2021), **Air Quality** (Cini et al., 2021), **Daily website visitors (DWV)**, **Daily Gold Price (DGC)**, and **Daily Demand Forecasting Orders (DDFO)**, **Bitcoin Historical Dataset (BTC)**. We also test RT's performance on two-time series classification tasks, i.e., **Absenteeism at work (AW)** and **Temperature Readings from IOT Devices** (details in the Appendix A.1). In our empirical evaluation, presented in Table 2, a comparison of multivariate long-term forecasting errors using the Mean Squared Error (MSE) metric was performed across various methods and datasets. For each dataset, a forecasting horizon was specified with ILI having horizons $T \in \{24, 36, 48, 60\}$ and the rest having $T \in \{96, 192, 336, 720\}$. It is evident that the **RT** consistently outperforms other methods, achieving the lowest MSE in most scenarios. The Transformer-based methods also show competitive performance, with some of their results being underscored for distinction (univariate time forecasting experiments in the Appendix A.5).

| Length/Dataset | ETTh1 | ETTh2 | ETTm1 | ETTm2 | Exchange | ILI | BTC | DGP | DWV | DDFO |
|---|---|---|---|---|---|---|---|---|---|---|
| 100 | 0.790 | **1.190** | 0.560 | 0.299 | 0.995 | 0.995 | **0.961** | 1.070 | 1.343 | **0.540** |
| 1000 | **1.340** | 0.973 | 0.931 | 0.535 | 1.099 | 1.249 | 0.853 | **1.626** | 1.823 | **0.540** |
| 10000 | 1.266 | 0.866 | **1.061** | **0.667** | 1.359 | 1.299 | 0.853 | 1.398 | **1.838** | **0.540** |

Table 1: This study investigates the correlation dimension $D_2$ across multiple datasets using diverse state numbers (100, 1000, and 10000). The correlation dimension $D_2$ serves as an indicator of chaotic behavior, with higher values denoting a more pronounced presence of chaos (Panis, 2020). Our findings indicate that the correlation dimension $D_2$ was evaluated for all datasets under varying state numbers.

**Test Chaotic Behavior:** To show that the datasets we have used are chaos, we employ the correlation dimension, denoted as $D_2$ serves as a reliable measure of chaotic behavior, with higher values reflecting more pronounced chaos. To evaluate the chaotic behavior of various time series datasets (Pánis et al., 2020), the $D_2$ is defined as

$$D_2 = \lim_{\epsilon \to 0} \frac{\ln C(\epsilon)}{\ln \epsilon}. \tag{15}$$

The correlation sum $C(\epsilon)$ for some small scalar $\epsilon$ is defined as in (Grassberger & Procaccia, 1983):

$$C(\epsilon) = \lim_{N \to \infty} \frac{2}{N(N-1)} \sum_{i<j} H(\epsilon - |x_i - x_j|), \tag{16}$$

where $H$ is the Heaviside step function, $N$ is the number of points, and $|x_i - x_j|$ is the distance between two points . We compute the correlation dimension for all datasets in our experiments using the implementation provided by (Vasquez-Correa, 2015)

Table 1 displays the outcomes of a correlation dimension ($D_2$) analysis across various datasets. The correlation dimension gauges chaotic tendencies, with higher values suggesting heightened chaos. This assessment involves different state numbers: 100, 1000, and 10000 and it exhibits $D_2$ values for each dataset and state number. The findings reveal that, for all datasets, the correlation dimension increases with higher state numbers, indicating intensified chaotic behavior.

## 4.2 ABLATION STUDY

**Lyapunov Exponent:** The Lyapunov Exponent (LE) measures how chaotic a system is by computing how quickly points on a path move away from each other over time. Table 3 shows our RT and

| Model | Horizons | Linear | NLinear | DLinear | FEDformer | Autoformer | Informer | Pyraformer | LogTrans | Repeat | RT |
|---|---|---|---|---|---|---|---|---|---|---|---|
| Electricity | 96 | **0.140** | 0.141 | **0.140** | 0.193 | 0.201 | 0.274 | 0.386 | 0.258 | 1.588 | **0.140** |
| | 192 | 0.153 | 0.154 | 0.153 | 0.201 | 0.222 | 0.296 | 0.386 | 0.266 | 1.595 | **0.151** |
| | 336 | 0.169 | 0.171 | 0.169 | 0.214 | 0.231 | 0.300 | 0.378 | 0.280 | 1.617 | **0.168** |
| | 720 | 0.203 | 0.210 | 0.203 | 0.246 | 0.254 | 0.373 | 0.376 | 0.283 | 1.647 | **0.201** |
| Exchange | 96 | 0.082 | 0.089 | **0.081** | 0.148 | 0.197 | 0.847 | 0.376 | 0.968 | **0.081** | 0.101 |
| | 192 | 0.167 | 0.180 | **0.157** | 0.271 | 0.300 | 1.204 | 1.748 | 1.040 | 0.167 | 0.192 |
| | 336 | 0.328 | 0.331 | **0.305** | 0.460 | 0.509 | 1.672 | 1.874 | 1.659 | **0.305** | 0.305 |
| | 720 | 0.964 | 1.033 | 0.643 | 1.195 | 1.447 | 2.478 | 1.943 | 1.941 | 0.823 | **0.586** |
| Traffic | 96 | 0.410 | 0.410 | 0.410 | 0.587 | 0.613 | 0.719 | 2.085 | 0.684 | 2.723 | **0.391** |
| | 192 | 0.423 | 0.423 | 0.423 | 0.604 | 0.616 | 0.696 | 0.867 | 0.685 | 2.756 | **0.399** |
| | 336 | 0.436 | 0.435 | 0.436 | 0.621 | 0.622 | 0.777 | 0.869 | 0.734 | 2.791 | **0.425** |
| | 720 | 0.466 | 0.464 | 0.466 | 0.626 | 0.660 | 0.864 | 0.881 | 0.717 | 2.811 | **0.442** |
| Weather | 96 | 0.176 | 0.182 | 0.176 | 0.217 | 0.266 | 0.300 | 0.896 | 0.458 | 0.259 | **0.173** |
| | 192 | 0.218 | 0.225 | 0.220 | 0.276 | 0.307 | 0.598 | 0.622 | 0.658 | 0.309 | **0.217** |
| | 336 | 0.262 | 0.271 | 0.265 | 0.339 | 0.359 | 0.578 | 0.739 | 0.797 | 0.377 | **0.259** |
| | 720 | 0.326 | 0.338 | 0.323 | 0.403 | 0.419 | 1.059 | 1.004 | 0.869 | 0.465 | **0.314** |
| ILI | 24 | 1.947 | 1.683 | 2.215 | 3.228 | 3.483 | 5.764 | 1.420 | 4.480 | 6.587 | **0.109** |
| | 36 | 2.182 | 1.703 | 1.963 | 2.670 | 3.103 | 4.755 | 7.394 | 4.799 | 7.130 | **0.109** |
| | 48 | 2.256 | 1.719 | 2.130 | 2.622 | 2.669 | 4.763 | 7.551 | 4.800 | 6.575 | **0.111** |
| | 60 | 2.390 | 1.819 | 2.368 | 2.857 | 2.770 | 5.264 | 7.662 | 5.278 | 5.893 | **0.113** |
| ETTh1 | 96 | 0.375 | **0.374** | 0.375 | 0.376 | 0.449 | 0.865 | 0.664 | 0.878 | 1.295 | 0.375 |
| | 192 | 0.418 | 0.408 | **0.405** | 0.420 | 0.500 | 1.008 | 0.790 | 1.037 | 1.325 | 0.411 |
| | 336 | 0.479 | **0.429** | 0.439 | 0.459 | 0.521 | 1.107 | 0.891 | 1.238 | 1.323 | 0.432 |
| | 720 | 0.624 | 0.440 | 0.472 | 0.506 | 0.514 | 1.181 | 0.963 | 1.135 | 1.339 | **0.436** |
| ETTh2 | 96 | 0.288 | 0.277 | 0.289 | 0.346 | 0.358 | 3.755 | 0.645 | 2.116 | 0.432 | **0.228** |
| | 192 | 0.377 | 0.344 | 0.383 | 0.429 | 0.456 | 5.602 | 0.788 | 4.315 | 0.534 | **0.269** |
| | 336 | 0.452 | 0.357 | 0.448 | 0.496 | 0.482 | 4.721 | 0.907 | 1.124 | 0.591 | **0.289** |
| | 720 | 0.698 | 0.394 | 0.605 | 0.463 | 0.515 | 3.647 | 0.963 | 3.188 | 0.588 | **0.306** |
| ETTm1 | 96 | 0.308 | 0.306 | **0.299** | 0.379 | 0.505 | 0.672 | 0.543 | 0.600 | 1.214 | 0.312 |
| | 192 | 0.340 | 0.349 | **0.335** | 0.426 | 0.553 | 0.795 | 0.557 | 0.837 | 1.261 | 0.348 |
| | 336 | 0.376 | 0.375 | **0.369** | 0.445 | 0.621 | 1.212 | 0.754 | 1.124 | 1.283 | 0.372 |
| | 720 | 0.440 | 0.433 | **0.425** | 0.543 | 0.671 | 1.166 | 0.908 | 1.153 | 1.319 | 0.424 |
| ETTm2 | 96 | 0.168 | 0.167 | 0.167 | 0.203 | 0.255 | 0.365 | 0.435 | 0.768 | 0.266 | **0.151** |
| | 192 | 0.232 | 0.221 | 0.224 | 0.269 | 0.281 | 0.533 | 0.730 | 0.989 | 0.340 | **0.187** |
| | 336 | 0.320 | 0.274 | 0.281 | 0.325 | 0.339 | 1.363 | 1.201 | 1.334 | 0.412 | **0.210** |
| | 720 | 0.413 | 0.368 | 0.397 | 0.421 | 0.433 | 3.379 | 3.625 | 3.048 | 0.521 | **0.285** |

Table 2: Comparing multivariate long-term forecasting errors using MSE: lower values indicate better performance. Specifically, for the ILI dataset, forecasting horizons are $T \in \{24, 36, 48, 60\}$, while for the remaining datasets, they are $T \in \{96, 192, 336, 720\}$. Optimal results are emphasized in **bold**, while the top results from Transformers are underlined for distinction."

| Horizons | | 100 | | 500 | | 1500 | | 2500 | |
|---|---|---|---|---|---|---|---|---|---|
| Dataset | Model | MSE | MAE | MSE | MAE | MSE | MAE | MSE | MAE |
| ETTh1 | NLinear | 5.017 | 1.992 | 3.144 | 1.628 | 14.247 | 2.924 | 16.550 | 3.327 |
| | RT | **1.640** | **1.413** | **1.159** | **0.968** | **7.753** | **2.002** | **9.601** | **2.361** |
| ETTh2 | NLinear | 17.428 | 5.476 | 39.197 | 6.841 | 109.833 | 8.751 | 101.251 | 8.462 |
| | RT | **8.793** | **1.770** | **14.163** | **2.334** | **19.161** | **2.935** | **48.803** | **4.821** |
| ETTm1 | NLinear | **0.678** | **1.539** | 9.036 | 2.395 | **9.428** | **2.476** | 11.153 | 2.768 |
| | RT | 0.943 | 2.072 | **7.057** | **2.075** | 9.549 | 2.492 | **10.837** | **2.562** |
| ETTm2 | NLinear | 49.464 | 5.802 | 53.319 | 5.720 | 60.884 | 6.348 | 78.999 | 7.260 |
| | RT | **12.740** | **2.707** | **18.027** | **3.029** | **22.740** | **3.707** | **44.180** | **5.136** |

Table 3: Comparison of NLinear and RT for explaining the Lyapunov Exponent's predictive performance across different datasets. The table presents MSE and MAE values for both models at various prediction horizons: 100, 500, 1500, and 2500. The results demonstrate that the RT model consistently outperforms the NLinear model, achieving significantly lower MSE and MAE values, thus overcoming the Lyapunov Exponent's challenges of far-sighted forecasting.

a baseline NLinear LE over different time steps of 100, 500, 1500, and 2500, in future on different datasets (ETTh1, ETTh2, ETTm1, ETTm2) and outperforms the NLinear model in predicting the LE. Figure 2 illustrates the comparison of forecasting 2500 future time steps between the NLinear and RT models, where RT matches more the ground truth than NLinear.

**Explain Signature Features:** We illustrate and measure how well-distributed the features input of Transformer with or without reservoir are. More distribution means higher entropy and more informative, and vice versa. We use the Local Interpretable Model-agnostic Explanations (LIME)

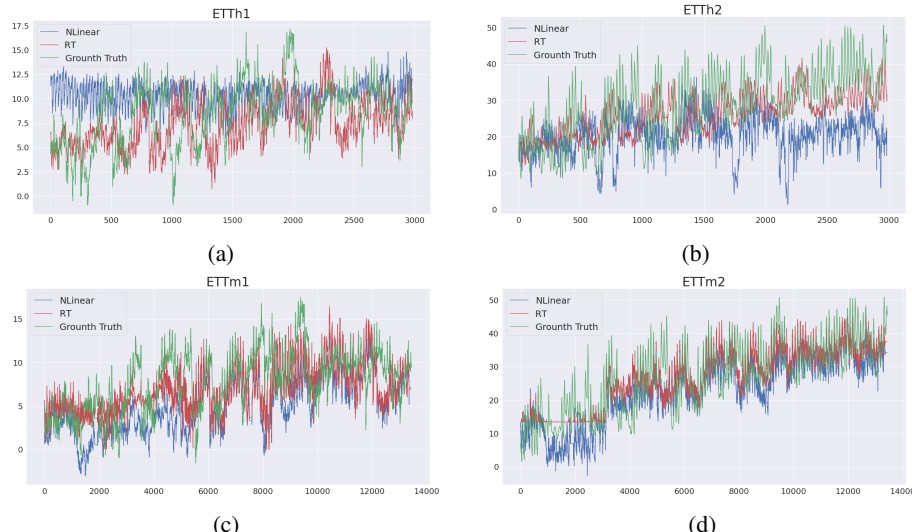

(a)

(b)

(c)

(d)

Figure 2: Visualization comparing the prediction of the Lyapunov Exponent using NLinear and RT models. The X-axis corresponds to the time sequence, while the Y-axis corresponds to the target feature.

Table 4: Top: Entropy in different datasets. This table presents the entropy values of the original inputs ($\vec{u}(t)$), the corresponding reservoir readouts ($\vec{o}(t)$), and the resultant transformed representations ($\vec{z}(t)$) across various datasets. Bottom: The Lime Explanation effect of current input (A) and reservoir (B) on output prediction is shown for ETTh1

Figure 3: In the presented graph, the X-axis represents the size of the model parameters in thousands, while the Y-axis illustrates the mean absolute error of the prediction outputs. The yellow boxes in the graph indicate the history lengths used in the computation of the RT-1R, RT-5R, and RT-10R models, which were each constructed with one, five, and ten reservoirs, respectively.

algorithm (Ribeiro et al., 2016). Our goal is to explain the decision-making process within the readout layer by employing LIME to generate a local linear approximation of the model. The initial step involves generating perturbed input time series by adding noise to the original $\vec{u}(t)$. A simpler interpretable model is then trained using these perturbed inputs along with corresponding output time series $\vec{y}(t)$. The complexity of the interpretable model depends on the reservoir model's intricacy, encompassing both linear and non-linear options. Ultimately, we utilize the coefficients of the interpretable model, along with LIME-calculated feature importances, to explain the readout layer's decision. This Table (Top) 4 shows that the readout from the reservoir, $\vec{o}(t)$, has higher entropy than Transformer original input features $\vec{u}(t)$. By concatenating both using Equation 11, we achieve the highest entropy. Additionally, Table (Bottom) 4 illustrates the Lime explanation for both RT and the baseline Transformer (more ablation in the Appendix A.4).

**History length versus parameter size:** Conventional approaches for modeling temporal data require a large number of training parameters, proportional to the window size (s) and the number of features (C). This becomes infeasible when considering long historical sequences. In contrast, reservoir computing approaches, such as the proposed method, only train a vector representation (i.e. readout) for the next time step without explicit conditioning on the entire history. Empirical results, illustrated in Figure 3, demonstrate that RT provides a better trade-off between the number of training parameters and loss.

## 5 RELATED WORK

Chaotic time-series prediction using machine learning approaches has been investigated in the last decade using CNN, WNN, FNN, LSTM, etc. (Ramadevi & Bingi, 2022). For example,(Sangiorgio & Dercole, 2020) proposed the use of a Recurrent Neural Network (RNN) with Long Short-Term Memory (LSTM) cells for chaotic time-series prediction and demonstrated improved performance on several chaotic systems.(Karasu & Altan, 2022) showed a new type of LSTM cell to handle chaotic time series prediction and reported that it outperforms traditional LSTM on several chaotic systems. Despite the domain knowledge studied in broad areas, developing new machine learning technologies for Chao's purpose still needs more attention. In particular, the connection between chaotic prediction and conventional time-series forecasting is yet to be more closely integrated (Sahoo et al., 2019). There has been extensive literature on time-series forecasting as well (PaperwithCodes, 2022), and sometimes these datasets hold chaotic properties as well, including using Transformer- and non-Transformer-based solutions for long-term time series forecasting (LTSF). For example, (Mohammadi Farsani & Pazouki, 2020) showed the use of self-attention in the Transformer model for time-series prediction tasks that improved performance. Similarly, (Huang et al., 2019) proposed the use of self-attention in a Recurrent Neural Network (RNN) architecture for time-series prediction, achieving improved performance on several datasets. However, these approaches still need more effective models for long-sequential input handling to consider the long history of the past. We have proposed a reservoir Transformer (RT) to reach this aim.

While reservoirs have found extensive application in time-series tasks, their performance often lags behind the current state-of-the-art outcomes (Gauthier, 2021). A concept of Reservoir Transformer (Shen et al., 2020) was introduced by interspersing the reservoir layers into the regular Transformer layers and showing improvements in wall-clock compute time. However, such integration will reduce the feature extraction power of both reservoirs in the long-sequence and Transformer in short sequences. We do not combine the reservoir and Transformer linearly but in a non-linear way and receive significant improvements over all previous methods in time-series prediction, including chaotic prediction.

Additionally, there has been research on using ensembles of models for time-series prediction.(Feng et al., 2019) proposed an ensemble method that combines multiple hybrid models using wavelet transform for chaotic time-series prediction and achieved improved performance on several chaotic systems.(Tang et al., 2020) proposed an ensemble method that combines multiple deep-learning models for time-series prediction and showed improved performance on several datasets. In our work, we apply ensemble techniques to our Reservoir Transformer for the first time, contributing to further advancements in time-series prediction.

## 6 CONCLUSION

We introduce nonlinear readout to reservoir computing that significantly enhances the performance of the state-of-the-art DNNs, including Transformer. Our deep Reservoir Transformer has the unique advantage of handling any input length and producing robust prediction outputs. Thus, it is ideal for chaotic time-series forecasting. This research will open new research questions on seamlessly integrating chaotic analysis, DNNs, and reservoir computing to better understand and predict our environments.

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

# A  APPENDIX

## A.1  DATASET AND PARAMETER CONFIGURATION

In this subsection, we outline the approach taken for dataset partitioning and parameter selection within our research framework.

**Dataset Partitioning:** The dataset is divided sequentially into three distinct sets: a training set, a validation set, and a test set. Specifically, the first 70% of the data is assigned to the training set, while 10% is allocated to the validation set, and the remaining 20% forms the test set. This partitioning strategy enables effective model training, validation, and evaluation.

**Model Parameters:** The configuration of model parameters plays a crucial role in our research methodology. For the Transformer architecture, we employ four Transformer blocks, each comprising 32 multi-head attention mechanisms with a total of four attentions. In the case of the Feedforward Neural Network (FNN), we utilize hidden layers containing 64 units each.

Within the network architecture, dropout is employed as a regularization technique. Specifically, a dropout rate of 0.4 is applied to the hidden layers, while a rate of 0.1 is used for the reservoir network. Additionally, dropout with a rate of 0.2 is integrated into the attention network.

Furthermore, the reservoir state size varies between 20 and 50 units across different ensembling reservoirs. To introduce diversity in the ensembling process, sparsity levels range from 0.3 to 0.7 for different reservoirs, accompanied by leaky units with values ranging from 0.2 to 0.6.

Finally, a learning rate of $1 \times 10^{-4}$ is adopted to facilitate the training process and achieve optimal convergence.

By meticulously configuring these parameters, we aim to attain a comprehensive understanding of the model's performance across various experimental setups.

**Evaluation metric:** In order to analyze the performance and evaluation of the proposed work, Mean Squared Error (MSE), and Mean Absolute Error (MAE) has been used for regression. The MSE is a quantification of the mean squared difference between the actual and predicted values. The minimum difference, the better the model's performance. On the other hand, MAE is a quantification of the mean difference between the actual and predicted values. As like as MSE, the minimum score of MAE, the better the model's performance. For classification tasks, we have used Accuracy and F1 score metrics.

**Dataset Description:**

- **ETTh (Electricity Transformer Temperature hourly-level):** The ETTh dataset presents a comprehensive collection of transformer temperature and load data, recorded at an hourly granularity, covering the period from July 2016 to July 2018. The dataset has been meticulously organized to encompass hourly measurements of seven vital attributes related to oil and load characteristics for electricity transformers. These incorporated features provide a valuable resource for researchers to glean insights into the temporal behavior of transformers.

  For the ETTh1 subset of the dataset, our model architecture entails the utilization of reservoir units ranging in size from 20 to 50, coupled with an ensemble comprising five distinct reservoirs. The attention mechanism deployed in this context employs a configuration of 15 attention heads, each possessing a dimension of 10 for both query and key components.

  Likewise, for the ETTh2 subset, our approach involves employing a total of seven ensemble reservoirs, each incorporating reservoir units spanning from 20 to 50 in size. The remaining parameters for this subset remain consistent with the configuration established for ETTh1.

- **ETTm (Electricity Transformer Temperature 15-minute-level):** The ETTm dataset offers a heightened level of detail and granularity by capturing measurements at intervals as fine as 15 minutes. Similar to the ETTh dataset, ETTm spans the timeframe from July 2016 to July 2018. This dataset retains the same seven pivotal oil and load attributes for electricity transformers as in the ETTh dataset; however, the key distinction lies in its higher temporal resolution. This heightened level of temporal granularity holds the potential to unveil subtler patterns and intricacies in the behavior of transformers.

In the context of model architecture, the configuration adopted for the ETTm dataset involves the use of ensemble reservoirs. Specifically, for the ETTm1 subset, an ensemble of seven reservoirs is employed, while for the ETTm2 subset, an ensemble of six reservoirs is utilized. This distinctive architectural setup is designed to harness the increased temporal resolution of the ETTm dataset, enabling the model to capture and interpret the finer nuances within the transformer data.

- **Exchange-Rate** This dataset includes the daily exchange rates for eight foreign countries: Australia, Britain, Canada, Switzerland, China, Japan, New Zealand, and Singapore. This data ranges from the year 1990 to 2016. To analyze this information, we are using 15 reservoirs, which are a specific type of model to help us understand and predict changes in these exchange rates.

- **AQI (Air Quality ):** The AQI dataset encompasses a collection of recordings pertaining to various air quality indices, sourced from 437 monitoring stations strategically positioned across 43 cities in China. Specifically, our analysis focuses solely on the PM2.5 pollutant for this study. Notably, previous research efforts in the realm of imputation have concentrated on a truncated rendition of this dataset, which comprises data from 36 sensors.

  For our investigation, we leverage 15 ensemble reservoirs as the architectural foundation for this dataset. This ensemble configuration is tailored to the distinctive characteristics and complexities inherent in the AQI data. By deploying this setup, we endeavor to enhance our capacity to effectively address the intricate imputation challenges associated with the AQI dataset.

- **Daily Website visitors:**[1] The dataset pertinent to daily website visitors provides a comprehensive record of time series data spanning five years. This dataset encapsulates diverse metrics of traffic associated with statistical forecasting teaching notes. The variables encompass daily counts encompassing page loads, unique visitors, first-time visitors, and returning visitors to an academic teaching notes website. The dataset comprises 2167 rows, covering the timeline from September 14, 2014, to August 19, 2020.

  From this array of variables, our specific focus centers on the 'first-time visitors,' which we have identified as the target variable within the context of a regression problem. By delving into this specific variable, we aim to uncover insights regarding the behavior of new visitors to the website and its associated dynamics.

  In terms of the model architecture employed for this dataset, we have constructed a configuration centered around 12 ensemble reservoirs. This architectural choice reflects the dataset's intricacies and the requirement for an approach that can effectively capture the underlying patterns and variations within the daily website visitor data

- **Daily Gold Price:** [2] The dataset pertaining to daily gold prices offers an extensive collection of time series data encompassing a span of five years. This dataset encapsulates a range of metrics pertaining to gold price dynamics. The variables included cover daily measurements related to various aspects of gold pricing. The dataset consists of 2167 rows, covering the time period from September 14, 2014, to August 19, 2020.

  The primary objective in this analysis is regression, wherein we seek to model relationships and predict outcomes based on the available variables. Specifically, we are interested in predicting the daily gold price based on the given set of features.

  In terms of the architectural setup for this dataset, we have established a model configuration that employs 10 ensemble reservoirs. This choice is rooted in the need to effectively capture the nuances and trends inherent in the daily gold price data. By employing this ensemble-based approach, we aim to enhance the model's capacity to discern patterns and fluctuations within the gold price time series.

- **Daily Demand Forecasting Orders:** [3] This is a regression dataset that was collected during 60 days, this is a real database of a Brazilian company of large logistics. Twelve predictive attributes and a target that is considered a non-urgent order. We used 17 ensemble reservoirs for this dataset.

---

[1] https://www.kaggle.com/datasets/bobnau/daily-website-visitors
[2] https://www.kaggle.com/datasets/nisargchodavadiya/daily-gold-price-20152021-time-series
[3] https://archive.ics.uci.edu/dataset/409/dailydemandforecastingorders

- **Bitcoin Historical Dataset (BTC):** [4] The dataset used in this research contains the historical price data of Bitcoin from its inception in 2009 to the present. It includes key metrics such as daily opening and closing prices, trading volumes, and market capitalizations. This comprehensive dataset captures Bitcoin's significant fluctuations and trends, reflecting major milestones and market influences. It serves as an essential tool for analyzing patterns, understanding price behaviors, and predicting future movements in the cryptocurrency market. We used 10 ensemble reservoirs for this dataset.

- **Absenteeism at Work:** [5] The dataset focusing on absenteeism at work is sourced from the UCI repository, specifically the "Absenteeism at work" dataset. In our analysis, the dataset has been utilized for classification purposes, specifically targeting the classification of individuals as social drinkers. The dataset comprises 21 attributes and encompasses a total of 740 records. These records pertain to instances where individuals were absent from work, and the data collection spans the period from January 2008 to December 2016.

  Within the scope of this analysis, the primary objective is classification, wherein we endeavor to categorize individuals as either social drinkers or non-social drinkers based on the provided attributes.

  The architectural arrangement for this dataset involves the utilization of 15 ensemble reservoirs. This choice of model configuration is underpinned by the aim to effectively capture the intricacies inherent in the absenteeism data and its relationship to social drinking behavior. Employing ensemble reservoirs empowers the model to discern nuanced patterns and interactions among the attributes, thereby enhancing its classification accuracy.

- **Temperature Readings: IOT Devices:** [6] The dataset pertaining to temperature readings from IoT devices centers on the recordings captured by these devices, both inside and outside an undisclosed room (referred to as the admin room). This dataset comprises temperature readings and is characterized by five distinct features. In total, the dataset encompasses 97605 samples.

  The primary objective in this analysis involves binary classification, where the aim is to classify temperature readings based on whether they originate from an IoT device installed inside the room or outside it. The target column holds the binary class labels indicating the origin of the temperature reading.

  In terms of dataset characteristics, the architecture of our model involves the utilization of 15 ensemble reservoirs. This architectural choice is rooted in the desire to effectively capture the underlying patterns and distinctions present within the temperature readings dataset. The use of ensemble reservoirs enhances the model's capacity to distinguish between the different temperature reading sources, thereby facilitating accurate classification.

## A.2 Pseudo Algorithm For Deep Reservoir Transformer

The proposed algorithm 1 is for training a deep reservoir computing model, with the goal of optimizing the distribution of the target variable, $\vec{y}(t+1)$, given the input variables, $\vec{u}(1:t+1)$, and previous target variables, $\vec{y}(1:t)$. The algorithm initializes the weights of the reservoir, $\vec{W}_{in}$ and $\hat{W}$, as well as the weights of the non-linear output, $\sigma$, and the Transformer encoder, $\phi$, as random variables sampled from a normal distribution with mean 0 and standard deviation 1.

The algorithm then enters an outer loop, where the number of iterations is determined by the number of epochs. Within this loop, there is an inner loop that iterates through each time step, $i$, from 1 to $T$. For each time step, a group of reservoirs, $GroupESN$, are processed in parallel. Each reservoir, $R_k$, receives the input, $\vec{u}(i)$, and the previous target variable, $\vec{y}(i)$, as input, and calculates a non-linear readout, $o^k(i)$, using the weights $\vec{W}_{in}, \hat{\vec{W}}$, and $\sigma$. The readouts from all reservoirs are then element-wise added together to produce $o(i)$.

The input of the next time step, $\vec{u}(i+1)$, is concatenated with the readout from the reservoirs, $o(i)$, using a parameter $\kappa$, to produce $\vec{z}(i+1)$. Finally, the Transformer encoder, $\mathcal{M}(.)$, is applied to $\vec{z}(i+1)$ using the parameters $\phi$, to produce the target variable for the next time step, $\vec{y}(1+1)$.

---

[4] https://finance.yahoo.com/quote/BTC-USD/history/

[5] https://archive.ics.uci.edu/ml/datasets/Absenteeism+at+work

[6] https://www.kaggle.com/datasets/atulanandjha/temperature-readings-iot-devices

---

**Algorithm 1** Training algorithm of Deep Reservoir Computing

---

**Require:** $\vec{u}(1:T, 1:c), \vec{y}(1:T)$
**Ensure:** Optimize $Pr(\vec{y}(t+1)|\vec{u}(1:t+1), \vec{y}(1:t))$ distribution by learning a model $M(.)$

0: $\vec{W}_{in}, \hat{\vec{W}} \leftarrow \mathcal{N}(0,1)$ {Weights initialization of Reservoir}
0: $\sigma \leftarrow \mathcal{N}(0,1)$ {Weights initialization of Reservoir non-linear output}
0: $\phi \leftarrow \mathcal{N}(0,1)$ {Weights initialization of Transformer encoder $M(.)$}
0: **while** $epoch < epochs$ **do**
0:    **for** $i \leftarrow 1$ to $T$ **do**
0:       **for** $k \leftarrow 1$ to $GroupESN$ **do**
0:          $o^k(i) \leftarrow R_k(\vec{u}(i), \vec{y}(i); \vec{W}_{in}, \hat{\vec{W}}, \sigma)$ {Non-linear readout from $k-th$ reservoir }
0:          $o(i) \leftarrow o(i) \oplus o^k(i)$ {Element-wise addition among all reservoirs }
0:       **end for**
0:       $\vec{z}(i+1) \leftarrow \kappa\vec{u}(i+1) + (1-\kappa)o(i+1)$ {Concatenation }
0:       $\bar{\vec{y}}(i+1) \leftarrow \mathcal{M}(\vec{z}(i+1); \phi)$
0:    **end for**
0:    $loss \leftarrow \mathcal{L}(\bar{\vec{y}}(i+1), \vec{y}(i+1))$ {Calculating loss for every time step $z$}
0:    $\phi, \sigma \leftarrow update$ {Update parameters}
0: **end while**=0

---

The loss is calculated for every time step, $z$, by comparing the predicted target variable, $\bar{\vec{y}}(1:T)$, with the true target variable, $\vec{y}(1:T)$, using a loss function, $\mathcal{L}(.)$. The parameters, $\phi$ and $\sigma$, are then updated to minimize the loss.

### A.3   VARIABLE DESCRIPTIONS

### A.4   ABLATION STUDY

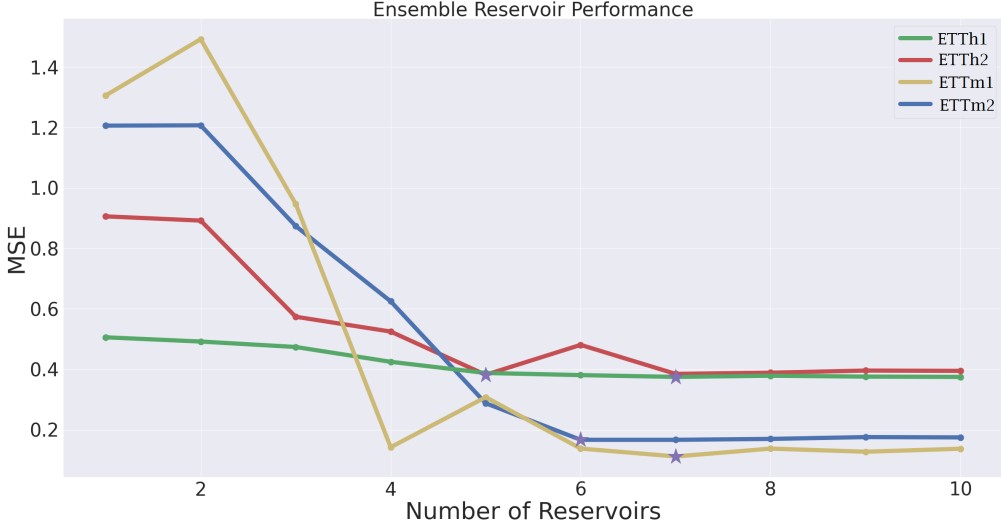

Figure 4: The graph illustrates the relationship between the number of ensemble reservoirs used and the mean square error (MSE) of the prediction outputs. The X-axis represents the number of reservoirs included in the ensemble, while the Y-axis depicts the corresponding MSE values. The 'star' in the graph denotes the minimum MSE achieved.

#### A.4.1   IMPACT OF RESERVOIR COUNT ON PERFORMANCE:

Figure 4 illustrates the relationship between the number of ensembling reservoirs and system performance. The x-axis depicts the count of reservoirs, while the y-axis portrays the Mean Squared Error

| Variable | Description |
|---|---|
| $\mathcal{M}(.)$ | A Transformer or DNN model. |
| $\vec{u}$ | Samples with $c$ components. |
| $c$ | Number of features in dataset |
| $\vec{y}$ | Target features for prediction. |
| $\vec{\hat{y}}$ | Prediction value |
| $T$ | Entire time span or duration |
| $t$ | Represents a specific moment or data point in time |
| $R$ | Non-linear readout representation of reservoir |
| $\vec{o}$ | Aggregate output from all reservoirs within the group |
| $\rho$ | Spectral radius of reservoir |
| $Q$ | Number of reservoirs in GroupESN |
| $m$ | Reservoir output size |
| $\vec{W}_i n$ | Input-to-reservoir weight matrix |
| $\vec{W}_{out}$ | Reservoir to readout weight matrix |
| $\theta_{out}$ | Reservoir to readout bias matrix |
| $\vec{\vec{W}}$ | Recurrent reservoir weight matrix |
| $\vec{x}$ | Reservoir state |
| $\kappa$ | Parameter to indicate the weights of current state and reservoir state |
| $\phi$ | Learnable parameters of $M(.)$ |
| $s$ | Window size to look back previous sequence |
| $R(.)$ | Deep reservoir computing |
| $d$ | Feature dimension of Transformer |
| $\vec{g}$ | Input for the baseline Transformer |
| $\vec{z}$ | The concatenation of reservoir and current multivariate |
| $\mathcal{L}$ | Loss function |
| $D_2$ | Correlation dimension |
| $C(\epsilon)$ | Correlation sum |
| $H(.)$ | Heaviside step function |

Table 5: Descriptions of all variables used in this paper

(MSE) value. The graph discernibly demonstrates that increasing the number of reservoirs positively correlates with performance enhancement, leading to a reduction in loss. However, beyond a certain threshold of reservoir count (best hyperparameters of the number of reservoirs), the performance improvement saturates, resulting in a plateau in loss reduction. This phenomenon suggests an optimal range for reservoir count in relation to performance enhancement.

### A.4.2 GRAPHICAL EXPLANATION OF FEATURES

In Figure 5, we present a visual depiction that offers a comprehensive insight into the interpretability of features derived from both the reservoir output $\vec{o}(t)$ and input feature $\vec{u}(t)$.

### A.5 MORE EXPERIMENTS

Table 6 shows that RT model outperforms GRIN (Cini et al., 2021), BRITS (Cao et al., 2018), ST-MVL (Yi et al., 2016), M-RNN (Yoon et al., 2018), ImputeTS (Moritz & Bartz-Beielstein, 2017) on the air quality dataset. On regression and classification tasks, RT consistently outperforms the baseline Transformer with respect to all criteria, achieving significantly lower MSE and MAE values across various datasets like DWV, DGP, DDFO, and BTC. This suggests that the RT model is better equipped to capture complex temporal patterns. In the classification tasks, the RT model demonstrates better performance in terms of accuracy, F1 Score, Jaccard Score, Roc Auc, Precision, and Recall across different datasets like Absenteeism at work and Temperature Readings. Through this analysis, we provide evidence that the RT architecture excels not only in regression tasks but also in classification tasks, outperforming traditional Transformer architectures.

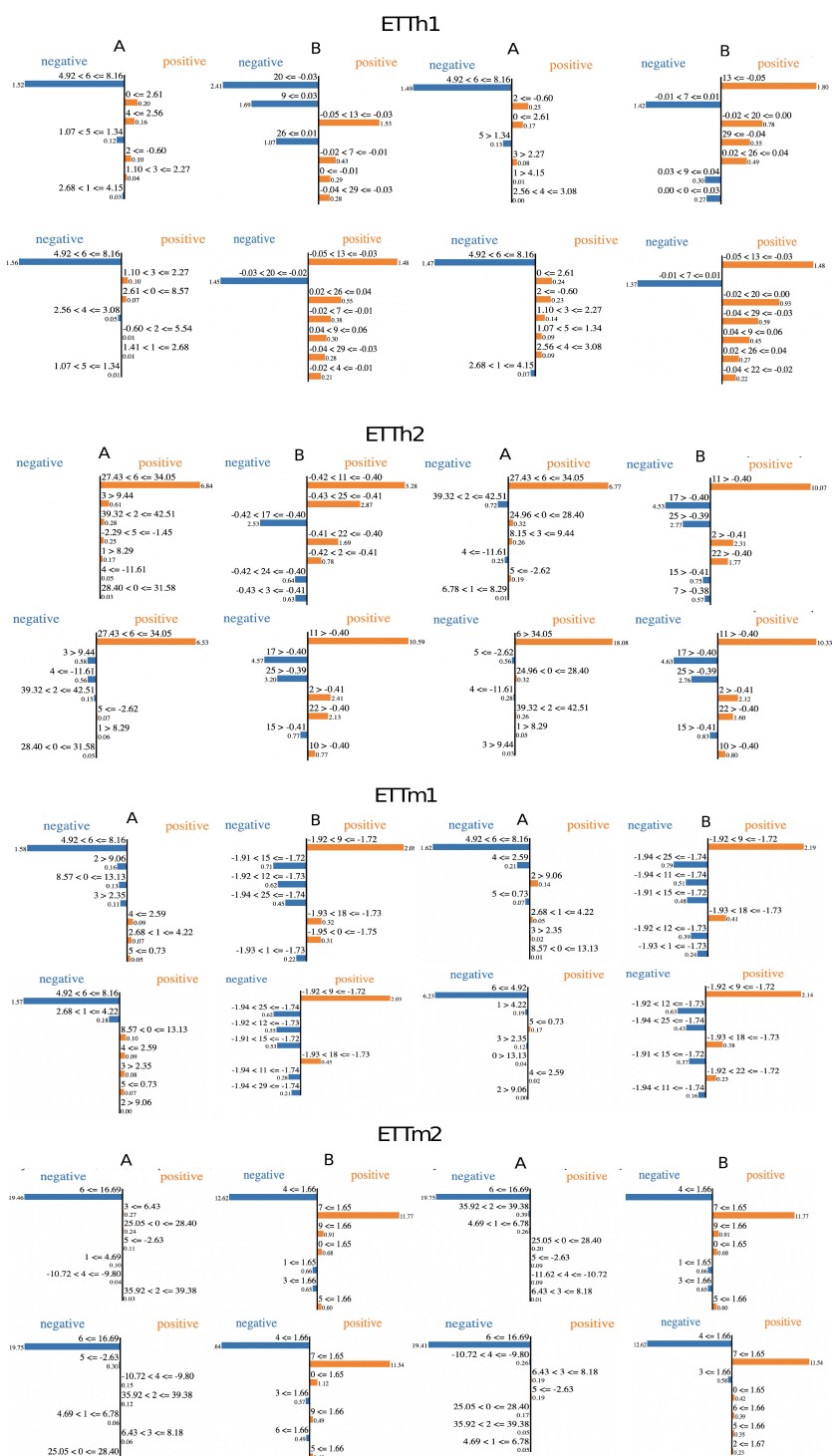

Figure 5: The Lime Explanation effect of current input (A) and reservoir (B) on output prediction is shown. The test dataset is represented by ETTh1, ETTh2, ETTm1, and ETTm2. The results indicate that current input has a greater impact on feature 6, while the reservoir affects multiple features such as 20, 13, 29, and 7.

| Air Quality | | | | | | | | |
|---|---|---|---|---|---|---|---|---|
| Metric/Model | GRIN | BRITS | ST-MVL | M-RNN | ImputeTS | RT | | |
| MAE | 10.514 | 12.242 | 12.124 | 14.242 | 19.584 | **9.046** | | |
| **Method** | | **Training** | | | | **Test** | | |
| **REGRESSION TASKS** | | | | | | | | |
| Model/Metric | MSE | MAE | R2 | MSE | MAE | R2 | Parameters | |
| Daily website visitors (DWV) | | | | | | | | |
| Baseline | 98.451 | 7.583 | 0.996 | 42.954 | 5.181 | 0.997 | ∼13.2k | |
| RT | **10.400 (-89.43%)** | 2.114 | 0.999 | 6.181 | 1.893 | 0.999 | ∼3.2k | |
| Daily Gold Price (DGP) | | | | | | | | |
| Baseline | 39459.790 | 112.251 | 0.996 | 168608.260 | 353.310 | 0.959 | ∼13.9k | |
| RT | **22022.981** | 86.985 | 0.998 | 28790.317 | 119.227 | 0.994 | ∼3.4k | |
| Daily Demand Forecasting Orders (DDFO) | | | | | | | | |
| Baseline | 312.933 | 13.274 | 0.929 | 1013.962 | 22.265 | 0.099 | ∼14.1k | |
| RT | **76.385** | 7.487 | 0.985 | 410.652 | 10.173 | 0.720 | ∼3.7k | |
| Bitcoin Historical Dataset (BTC) | | | | | | | | |
| Baseline | 188614.931 | 427.801 | 0.968 | 79389.491 | 232.119 | 0.961 | ∼10.5k | |
| RT | **158651.081** | 310.918 | 0.998 | 74800.535 | 211.626 | 0.989 | ∼2.8k | |
| **CLASSIFICATION TASKS:** | | | | | | | | |
| Model | Accuracy | F1 Score | Jaccard Score | Roc Auc | Precision | Recall | Parameters | |
| Absenteeism at work | | | | | | | | |
| Baseline | 0.928 | 0.940 | 0.887 | 0.929 | 0.953 | 0.927 | ∼14.2k | Training |
| RT | **0.947** | 0.956 | 0.916 | 0.950 | 0.976 | 0.936 | ∼3.5k | Training |
| Baseline | 0.844 | 0.839 | 0.722 | 0.844 | 0.833 | 0.845 | ∼14.2k | Test |
| RT | **0.899** | 0.901 | 0.819 | 0.901 | 0.944 | 0.860 | ∼3.5k | Test |
| Temperature Readings | | | | | | | | |
| Baseline | 0.888 | 0.934 | 0.877 | 0.877 | 0.877 | 0.915 | ∼10.9k | Training |
| RT | **0.959** | 0.975 | 0.952 | 0.934 | 0.981 | 0.970 | ∼2.3k | Training |
| Baseline | 0.649 | 0.688 | 0.524 | 0.646 | 0.632 | 0.755 | ∼10.9k | Test |
| RT | **0.802** | 0.843 | 0.728 | 0.794 | 0.867 | 0.820 | ∼2.3k | Test |

Table 6: Assessing the performance of regression on both training and test datasets, we utilize a window size of $s = 5$ to observe historical data in baseline Transformers.

