# OpenReview forum: "Reservoir Transformer at Infinite Horizon: the Lyapunov Time and the Butterfly Effect"
_ICLR.cc/2024/Conference — Submitted to ICLR 2024_

### Official Review · Reviewer_Md14 · 2023-10-20

**Soundness:** 3 good
**Presentation:** 3 good
**Contribution:** 2 fair
**Rating:** 6
**Confidence:** 2

**Summary:**

The manuscript proposes a transformer architecture for chaotic time series prediction, in particular addressing the challenge of supplying very long input sequences by routing the inputs first though an ensemble of (fixed weight) reservoir networks, from which the transformer then takes a lower-dimensional, non-linear readout. Performance comparisons to related models for chaotic time series prediction are shown.

Unfortunately I could not perform a suitably deep review of this manuscript due to time constraints. I apologize to the authors and the area chair. My comments are to be considered as low-confidence opinions.

**Strengths:**

Mostly clear presentation, promising empirical validation results

**Weaknesses:**

(see questions)

**Questions:**

- From a theoretical standpoint, it is not obvious to me why using reservoir networks should be more effective than employing other fixed nonlinear feature maps which map from an infinite time-horizon to a finite-dimensional state and might be computationally cheaper. In order to demonstrate that reservoir RNNs are specifically well suited, it would be useful to compare to several other input maps, while keeping the transformer architecture on top the same.
- Could it be useful to consider also some well-controlled synthetic time-series prediction tasks such as the dynamics of the Mackey-Glass equation or the Kuramoto-Shivashinsky system? In these tasks the Lyapunov time and dimensionality (in the KS case) can be varied in a controlled manner, facilitating a more detailed analysis of the model predictions and performance.
- Maybe some of the statements are overly strong.. (e.g. "significant departure from heuristic based assumptions", "transformative phase for the transformer model")

---

### Official Review · Reviewer_8rjM · 2023-10-30

**Soundness:** 2 fair
**Presentation:** 2 fair
**Contribution:** 2 fair
**Rating:** 3
**Confidence:** 3

**Summary:**

Authors, present resevoir transformers, which modify the original echo state networks (ESN), to include self-attention in the readout layer, an ensemble of reservoirs and a transformer that takes as input the processed reservoir state $z$ to produce the output $y$. Unlike reservoir transformers (Shen et.al (2020)), they endow existing ESN with transformers, instead of imposing the same training paradigm of ESN on transformers.

Their method is benchmarked against state-of-the-art transformer architectures for time-series forecasting in both regression and classification settings, where they show superior performance. Additionally, they study their model's performance by predicting the lyapunov exponent which is characteristic of a dynamical system. Finally, LIME is used to explain the readout layer's decision.

**Strengths:**

- Competitive performance against time-series baselines on standard benchmarks
- Outperforms NLinear on very long sequence forecasting
- Motivates the use of datasets by assessing their chaotic behavior
- Used LIME to explain readout layer's decision
- Good motivation of the resevoir ensemble, as showcased in the sensitivity analysis, in the appendix section

**Weaknesses:**

- The use of the transformer (and self-attention in general) is not very well motivated :
   - In section 3.1, you want to replace the linear readout by a transformer stating that for large $m$, the transformer training is slow, yet, in the previous section, you mention that for a linear readout, ESN is optimized using linear regression which is less costly and easier to perform, so why replace it with self-attention? A two-layer MLP could be enough since the self-attention doesn't act on a sequence but a vector in this case
   - The use of transformer after (that is on $z(t+1)$) is unclear. It is used on which sequence? From the algorithm in the appendix, the transformer is applied on $z(i+1)$ at each timestep $i$, so it's a vector not a sequence of vectors. Unless you mean the transformer is applied on a sequence $z(1), z(2), \dots, z(i+1)$ in which case, you should clarify that in the paper.
- Apart from the reservoir ensemble study and related to the above point, no ablation study is performed to justify the changes made to the original ESN. Hence, self-attention and transformer should be ablated separately as well to assess their contributions.
- You present a method that improves over vanilla ESN, yet, no ESN-derived baselines are included. You should at least compare against a vanilla ESN and the reservoir transformer (which is the closest to your method) (Shen, et.al (2020)).
-

**Questions:**

- In eq. 3, there's an $a$, perhaps you meant $\alpha$?
- I don't understand on what time-series does the transformer act on ? It takes $z(t+1)$ as input but that's just a single vector.
- Is the lyapunov exponent given for each point in the timeseries? My original understanding of lyapunov exponents is that it's computed over the whole sequence, so Figure 2 is (as well as the whole LE experiment) is unclear to me.

---

### Official Review · Reviewer_cN3c · 2023-10-31

**Soundness:** 2 fair
**Presentation:** 2 fair
**Contribution:** 2 fair
**Rating:** 3
**Confidence:** 2

**Summary:**

This paper combines Deep Reservoir Computing and transformers for long time series prediction, which can present chaotic behaviors. The method can handle arbitrarily input length sequences. The architecture consists of an ensemble of reservoirs, which readout is non-linear and modeled by attention mechanisms, thus allowing large expressive power. The ensembling technique is used to improve performances but also to limit the impact of sensitivity to initial conditions. The experiments include long-term time series forecasting on various datasets as well chaotic time series forecasting. Additionally, LIME algorithm is used to have some degree of post-processing interpretability.

**Strengths:**

- The method is interesting and novel, combining the expressive power of transformers with the properties of reservoir computing.
- The model is able to handle efficiently long input length sequences.
- The experimental results at very-long term forecasting on the ETT dataset are interesting, showing that the model manages to overall capture the trend, even though variations are still not well captured. This shows interesting improvement compared to the NLinear baseline.

**Weaknesses:**

The presentation of the paper could be better for three reasons:

- The message is not always clear. Indeed, studying the butterfly effect would include analysis on the sensitivity to initial conditions and study of completely chaotic systems such as the Lorentz equation. Moreover, one of the main tables of results consists of long-term time series forecasting, not really highlighting the ability to predict chaotic time series. This is interesting to have but is presented as a main argument whereas, in my opinion, it should be less important. This does not help clarifying the message.
- The details of the model and Deep Reservoir Computing are not always clearly explained, thus leading less experienced readers to go to external resources to really understand the basic concepts.
- There is plenty of notations problems and typos, which does not help the reading and lower the quality of the presentation. For instance in Eq.3, it should probably be $\alpha a$ instead of $a$, and $a$ is not defined, notations on the look-back window and the look-back window size are the same, $s$,  the phrase at the end of page 5, a sentence is not finished, etc...

The experiments are not completely extensive:
- State-of-the-art transformer for time series forecasting, PatchTST [1], is not included in the related work nor the experimental baselines, where it should be.
- The size of the look-back window for the experiments is not specified, which is crucial for baseline performances in long-term time series forecasting.
- More experiments on chaotic time series could be undertaken, since this is the main point of the article. All the baselines could be implemented for Table 3 and other the other datasets used in Table 2 could be used all well for Table 3. Especially since Table 3 presents the same underlying phenomena, whereas the other datasets present other features.

 ### Remarks:
When describing the basic Leaky Integrator ESN, [2] should be cited, since this is the main reference for this architecture.

Figure 1 is not very clear. It would probably be beneficial to combine the two ideas and have a single figure showing both the non-linear readout and the ensembling. As in the article, the link between Figure 1a and Figure 1b is not explicit, thus the figure is not self-explained.


[1]: Nie, Y., Nguyen, N. H., Sinthong, P., Kalagnanam, J. (2022). A time series is worth 64 words: Long-term forecasting with transformers. arXiv preprint arXiv:2211.14730.

[2]: Jaeger, H., Lukoševičius, M., Popovici, D., Siewert, U. (2007). Optimization and applications of echo state networks with leaky-integrator neurons. Neural networks, 20(3), 335-352.

**Questions:**

- How did you choose the different model parameters ? If a grid search was performed, it could be interesting to show a few ablation studies to better understand the training of the model and which choices are important for its performances.
- In Eq. 12, have you tried transforming the output $z$ with something else than a transformer ?
- Is it possible to perform a meaningful analysis of the significant features that influence model choices when applying LIME interpretation? Or do the features have no meaning or possible interpretation in the ETT case?

---

### Official Review · Reviewer_rA4y · 2023-11-10

**Soundness:** 3 good
**Presentation:** 3 good
**Contribution:** 2 fair
**Rating:** 5
**Confidence:** 4

**Summary:**

The work proposes a Reservoir Transformer network for time series forecasting specifically targeted towards arbitrarily long input sequences such that it could predict "chaotic" time series (i.e with positive Lyapunov exponents). The proposed network achieves this by processing the arbitrarily long input by a reservoir network with nonlinear output implemented by self-attention mechanism. The output of the reservoir network along with current time is passed to a transformer. To increase effectiveness of the approach an ensemble of reservoir networks is trained in parallel and their nonlinear output is then merged. The approach is tested on multiple (10) time series prediction tasks.

**Strengths:**

1. The work proposes a novel approach of extracting "historical" features from time series by reservoir network which then further can be used to transfer that information to a time-series transformer. This allows the transformer to consider arbitrary long inputs.

2. The approach proposes a novel nonlinear readout in terms of self-attention for the reservoir network.

3. A possibility of merging features from multiple reservoir networks to enhance prediction effectiveness is presented.

**Weaknesses:**

1. The approach accuracy at forecasting is compared with other transformer configurations, however does not compare with other network architectures designed for sequences such as RNNs and their variants (while M-RNNs are listed quantitive comparison is not presented), Graph Neural Networks, etc.

2. The reason for the choice of reservoir computing as feature extractor is not explained and ablations studies with other potential extractors, eg. auto encoders have not been considered. Similarly, the choice of self-attention output vs other projections have not been investigated.

3. Figure 3 and compared models are not well explained in the text.

**Questions:**

1. How would the approach apply to non-chaotic time-series and what would be the accuracy then? In other words, is chaos required for the forecasting?

2. Would the approach apply to solve chaotic differential equations?

3. Force Learning and its extensions (Full Force, R-FORCE) was proposed for optimization of Echo state networks. Would such optimization be applicable for RT?

---

### Meta-Review · Area_Chair_1Qwv · 2023-12-05

**Metareview:**

This paper introduces Reservoir Transformers, designed to model long-context time series problems, particularly those with chaotic behavior. The reviewers appreciated many aspects of the paper, including the ability to combine the expressive power of transformers with the benefits of reservoir computing. However, they also found that the approach is somewhat lacking in motivation, the baseline comparisons incomplete, and the utility of the approach questionable in comparison to other methods.

**Justification For Why Not Higher Score:**

The architecture lacks sufficient motivation and the experimental results are not convincing.

**Justification For Why Not Lower Score:**

N/A

---

### Decision · Program_Chairs · 2024-01-16

Reject